# Radiomics Analysis in Ovarian Cancer: A Narrative Review

Francesca Arezzo [1,*] , Vera Loizzi [2] , Daniele La Forgia [3] , Marco Moschetta [4] , Alberto Stefano Tagliafico [5] , Viviana Cataldo [1] , Adam Abdulwakil Kawosha [6] , Vincenzo Venerito [7] , Gerardo Cazzato [8] , Giuseppe Ingravallo [8] , Leonardo Resta [8] , Ettore Cicinelli [1] and Gennaro Cormio [1]

1. Obstetrics and Gynecology Unit, Department of Biomedical Sciences and Human Oncology, University of Bari "Aldo Moro", Piazza Giulio Cesare 11, 70124 Bari, Italy; viviana.cataldo7@gmail.com (V.C.); ettore.cicinelli@uniba.it (E.C.); gennaro.cormio@uniba.it (G.C.)
2. Obstetrics and Gynecology Unit, Interdisciplinar Department of Medicine, University of Bari "Aldo Moro", Piazza Giulio Cesare 11, 70124 Bari, Italy; vera.loizzi@uniba.it
3. SSD Radiodiagnostica Senologica, IRCCS Istituto Tumori Giovanni Paolo II, Via Orazio Flacco 65, 70124 Bari, Italy; d.laforgia@oncologico.bari.it
4. Breast Care Unit, Department of Emergency and Organ Transplantation, University of Bari "Aldo Moro", Piazza Giulio Cesare 11, 70124 Bari, Italy; marco.moschetta@uniba.it
5. Radiology Section, Department of Health Sciences (DISSAL), University of Genova, Via L.B. Alberti 2, 16132 Genoa, Italy; alberto.tagliafico@unige.it
6. Department of General Medicine, Universitatea Medicina si Farmacie Grigore T Popa, Strada Universitatii 16, 700115 Iasi, Romania; adam.akawosha@gmail.com
7. Rheumatology Unit, Department of Emergency and Organ Transplantations, University of Bari "Aldo Moro", Piazza Giulio Cesare 11, 70124 Bari, Italy; vincenzo.venerito@uniba.it
8. Pathology Section, Department of Emergency and Organ Transplantation, University of Bari "Aldo Moro", Piazza Giulio Cesare 11, 70124 Bari, Italy; gerycazzato@hotmail.it (G.C.); giuseppe.ingravallo@uniba.it (G.I.); leonardo.resta@uniba.it (L.R.)
* Correspondence: francescaarezzo@libero.it; Tel.: +39-327-496-1788

**Abstract:** Ovarian cancer (OC) is the second most common gynecological malignancy, accounting for about 14,000 deaths in 2020 in the US. The recognition of tools for proper screening, early diagnosis, and prognosis of OC is still lagging. The application of methods such as radiomics to medical images such as ultrasound scan (US), computed tomography (CT), magnetic resonance imaging (MRI), or positron emission tomography (PET) in OC may help to realize so-called "precision medicine" by developing new quantification metrics linking qualitative and/or quantitative data imaging to achieve clinical diagnostic endpoints. This narrative review aims to summarize the applications of radiomics as a support in the management of a complex pathology such as ovarian cancer. We give an insight into the current evidence on radiomics applied to different imaging methods.

**Keywords:** ovarian cancer; radiomics; machine learning; precision medicine

## 1. Introduction

Recently, there has been increased interest in AI techniques applied to radiomic analysis. These could be applied to medical imaging with the aim of developing automated tools that aid the detection and characterization of neoplasms [1].

Radiomic analysis also transforms large numbers of quantitative imaging features (i.e., shape, texture, signal intensity, and wavelength features) into mineable data by machine learning tools [2–5]. Such information could be applied in creating descriptive and predictive models that are capable of providing diagnosis and prognosis in different tumors and serve as useful decision support tools [6].

Recent studies in oncology have shown the ability of radiomics in the study of various tumor classification and staging methods [7], the prediction of biological features of tumors [8], the risk of lymph node metastasis [9], disease-free survival rates [10], reoccurrence risk [11], neoadjuvant chemotherapy [12], and chemoradiation response [13].

Quantitative imaging based on radiomics allows for inter- and intra-observer variability to be reduced in studies based on visual and qualitative observation [14–16].

## 2. How Is a Radiomic Analysis Performed?

The radiomic analysis process is carried out in different steps. The first step is image acquisition and segmentation. Then, the process continues with feature extraction and selection (these features are used to realize a model), followed by the final step which involves performance assessment of the algorithm.

It could be applied to many imaging techniques such as X-ray, ultrasound scan (US), computed tomography (CT), magnetic resonance imaging (MRI), or positron emission tomography (PET). To begin, a field of interest (ROI) or volume of interest (VOI), as well as the dedicated area, must be outlined by humans or machines. To save time, as well as cost and subjectivity or error related to manual segmentation, clearly defined algorithms that are able to segment automatic or semi-automatic lesions have been developed [17]. Sometimes, the compound nature of the medical images (ill-defined borders of the lesion, changes in the form of tissues after chemotherapy or radiotherapy, recognizability of the lesion from adjacent anatomical structures) may limit the performance of these algorithms.

In such situations, the segmentation and delineation of the ROI performed manually by an expert on one selected 2D slice or the entire 3D volume could provide the solution [18,19].

### 2.1. Feature Extraction

The feature extraction involves the use of dedicated software or programming packages such as 3DSlicer or Pyradiomics [20]. In order to customize the feature extraction and allow the use of more filters on original images, this step may include pre-processing equipment, such as discretization, resampling, and normalization.

These features can be categorized as follows:

— Statistical features;
— Shape-based features;
— Textural features.

Generally, the number of features extracted could be from 50 to 5000. This number decreases due to the selection of reproducible and informative characteristics and to prevent over-fitting, which is building an analytical process correlating closely to a specific dataset, with the scarce capability of generalizing on independent data [21].

### 2.2. Feature Selection

Feature selection involves removing inessential features and checking inter- and intra-observer feature variability to remove highly variable ones [22]. Hence, only features with a high similarity are selected. No consensus on the number of maximum selected features has been agreed on till now; this number might differ based on the dataset and data analyst [23].

### 2.3. Model Construction

The aim is to create a model related to a single outcome of interest (diagnosis, tumor response, patient outcome, etc.). When quantitative imaging features are applied to this algorithm, it analyzes the training data and works out a hypothesis to predict a variable.

The most known algorithms are random forest (RF) [24], least absolute shrinkage and selection operator (LASSO) [25], artificial neural networks (ANNs) [26], support vector machine (SVM) [27], and minimum redundancy maximum relevance (mRMR) [28]. The aforementioned are not meant to be exhaustive, and there is no superior algorithm; one algorithm might work better than another depending on the dataset type, attributes, and hyperparameter tuning [29].

### 2.4. Performance Assessment of the Algorithm

The performance of the algorithms can be evaluated with the area under receiver operating characteristic curve (AUROC).

Model validation can be carried out internally or externally. A well-known technique for internal validation is the leave-one-out cross-validation (LOOCV) in which all the data are used for training except for the one data point left out for validation and testing.

The bootstrap is another strategy whereby large series of data are generated (bootstrap sample) to represent a patient chosen at random, in combination with their corresponding characteristics and outcome. The process is then made again for the whole cohort of patients [30].

However, external validation of an independent prospective cohort remains the most trusted method for the testing of prognostic algorithms.

This paper aims to analyze the application of radiomics to different imaging techniques in the study of ovarian cancer, starting by reviewing the role of radiomics in ultrasound, then in MRI, and finally in CT.

### 3. Ovarian Cancer

Ovarian cancer (OC) ranks as the seventh most diagnosed cancer among women worldwide and the second most common gynecological malignancy. It was responsible for about 14,000 deaths in 2020 in the US [31].

About 90% of cancers of the ovary are epithelial ovarian cancer (EOC) types. OC has multiple cellular origins. The term tubo-ovarian cancer is often used because OC may manifest as an ovarian or fallopian tube mass or as a primary peritoneal cancer [32].

Type I cancers (low-grade serous, mucinous, endometrioid, and clear cell) arising from the ovaries are typically less aggressive and very likely to be diagnosed promptly, as they usually grow slowly. Type II malignancies (high-grade serous carcinomas (HGSC), undifferentiated carcinomas, and carcinosarcomas) may develop from the tubal and/or ovarian surface epithelium and are usually more aggressive [33,34].

The absence of proper screening and diagnostic programs capable of detecting OC at the early stage and the fast spread of disease through the surface lining of the peritoneum represent the main factors causing OC lethality [35]. There is still a lack of an accurate protocol to identify high-risk patients, and the identification of tools for accurate screening, early diagnosis, and prognosis of OC represents a currently unmet clinical need.

Furthermore, the role of medical imaging (US, RMI, CT, and PET) in OC is fast evolving. Soon, a simple tumor description and its extension may not be sufficient. The use of precision medicine may help to answer questions related to treatment response, best timing for surgery, prognosis, or molecularly targeted drug therapy [36,37].

Hence, radiomic technology has been introduced as an emergent tool for post-processing of medical images and the advancement of new quantification metrics linking qualitative and/or quantitative imaging data to clinical results [38–41].

### 4. Methods

In this narrative review, we extensively searched Medline and Scopus for manuscripts describing the deployment of radiomic analysis in ovarian cancer patients from January 2017 to January 2021. In particular, "radiomics" or "radiomic analysis" plus "cancer, ovarian" (an MeSH term) were searched as keywords. Any research study type was retrieved, with focus on those reporting radiomic features as part of predictive models for ovarian cancer clinical outcomes.

### 5. Ultrasound

US is a cheap, easily accessible, and well-recognized image modality for diagnosing and assessing OC; it is a non-invasive exam without radiation [42].

To facilitate the sonographer's assessment to differentiate benign from malignant adnexal masses, the International Ovarian Tumor Analysis group developed the Assessment

of Different Neoplasia in the Adnexa (ADNEX) model [43–48]. The Society of Radiologists in Ultrasound consensus statement [49] and the Gynecologic Imaging Reporting and Data System, also known as GI-RADS [50], are other proposed characterization and management systems of ovarian masses (OM). A standardized lexicon including all appropriate descriptors of the characteristic US appearance of normal ovaries and lesions of the ovaries was made available by the Ovarian-Adnexal Reporting and Data System (O-RADS), published in 2018. In 2020, based on this reporting system, US guidelines for management of lesions related to the ovaries were proposed, including all risks and strategies that correspond to their management [51].

However, about 18% to 31% of such lesions of the adnexa detected by ultrasound are unknown.

Radiomics has emerged as a new powerful method that is capable of quantifying features from ultrasound images [52,53]; these features contain information that reflects the underlying pathophysiology of a cancer tissue [54,55].

We report ultrasound studies deploying radiomic analysis and their technical characteristics (Table 1).

**Table 1.** Technical characteristics of reviewed ultrasound studies deploying radiomic analysis.

| Author | Year | Population Number | Radiomic Platform | Radiomic Features | ML Algorithms | Outcome | Study Design |
|---|---|---|---|---|---|---|---|
| Chiappa et al. [42] | 2020 | 241 | TRACE4© | 319 | Ensemble | Benign/malignant Oms | Retrospective |
| Jin et al. [56] | 2021 | 127 | Python 3.7.0 and package Pyradiomics 2.2.0 | 97 | U-net | Segmentation (U-net vs. human) | Retrospective |

OMs, ovarian masses.

Chiappa et al. aimed to test the performance of radiomic analysis in automatically distinguishing malignant and benign OMs on patients diagnosed with OMs.

They analyzed three types of OMs—solid, cystic, and mixed. The best accuracy was reported for cystic OMs with an 87% accuracy compared to 80% for solid OMs and 81% for mixed OMs.

Therefore, the application of radiomics to ultrasonography might improve the diagnosis of OC, and the main advantage of this algorithm is that it does not depend on the experience of the ultrasound examiner [42].

In another study, Jin et al. focused on the segmentation accuracy and its effect on radiomic features compared to manual segmentation performed by expert radiologists.

They applied multiple U-net models to realize automatic segmentation of target volumes for OC on US images.

U-Net architecture is a convolutional neural network that was developed for biomedical image segmentation and consists of a contracting path to capture context and a symmetric expanding path that enables precise localization [57].

These models achieved a relatively high accuracy on target representation with a high Pearson correlation (0.87, 95% CI 0.84–0.90) and intraclass correlation coefficients (0.85, 95% CI 0.82–0.88) in correlation with features extracted from manual contours [56].

## 6. Magnetic Resonance Imaging

The value of MRI in the assessment of adnexal masses has increased over time [58–60].

With high soft-tissue resolution and no radiation, MRI is widely used to evaluate the etiology of adnexal lesions. Despite these aforementioned advantages, it is still not easy to identify OC subtypes with MRI alone [61–64], and the differentiation of masses remains a challenge due to the morphological complexity of tumors and overlap [65].

Ovarian-Adnexal Reporting and Data System Magnetic Resonance Imaging (O-RADS MRI), similar to the aforementioned O-RADS ultrasound, was also recently proposed. It

aims to ensure uniform clear MRI evaluations of ovarian or other adnexal lesions, giving each lesion a risk category of malignancy, which directs the proper management. Five risk classes were established [66]. The latter algorithm was validated prospectively in a study on 1340 women. The AUROC for differentiating benign from malignant tumors was 0.961 (95% CI 0.948–0.971) among skilled radiologists [67].

According to many studies encountered, the qualitative diagnosis of OC by MRI is superior to other imaging methods [68–70]; nevertheless, more advanced techniques such as radiomics analysis may be necessary to improve the characterization and/or differentiation of complex adnexal masses by analyzing additional aspects as reported in Table 2 [71].

**Table 2.** Technical characteristics of reviewed MRI studies deploying radiomic analysis.

| Author | Year | Population Number | Radiomic Platform | Radiomics Features | ML Algorithms | Outcome | Study Design |
|---|---|---|---|---|---|---|---|
| Zhang et al. [72] | 2018 | 286 | MATLAB software | 1714 | LASSO | Benign/malignant OMs Type I/type II subtypes | Retrospective |
| Song et al. [73] | 2020 | 104 | ITK-SNAP software (version 4.7.2) | 960 | Logistic regression | Benign/borderline/ malignant OMs | Prospective |
| Jian et al. [74] | 2020 | 294 | ITK-SNAP software (version 4.7.2) | 851 | LASSO | Type I/type II subtypes | Retrospective |

OMs, ovarian masses; LASSO, least absolute shrinkage and selection operator.

Zhang et al. assessed the ability of radiomic MRI to classify OMs in benign or malignant and in type I and type II OC. They also determined the association between MRI radiomics and survival among OC patients. Their results reported that radiomic features derived from MRI were highly correlated with OC classification and prognosis with high accuracy. In particular, for benign versus malignant classification, the accuracy was 0.90, and for the type I and type II OC discrimination task, the accuracy was 0.93 [72].

Furthermore, to differentiate benign, malignant, and borderline OMs, Song et al. evaluated the ability of second and third class classification predictive tasks done with radiomic features applied to a dynamic contrast-enhanced magnetic resonance imaging (DCE-MRI) pharmacokinetic (PK) protocol.

The second class classification task was split into three subtasks—benign vs. borderline (task A), benign vs. malignant (task B), and borderline vs. malignant (task C).

The combination of the PK radiomics signatures model (PK model) represents a potential technique to carry out a differential diagnosis among benign, borderline, and malignant OM [73], showing an AUC of 0.899, 0.865, and 0.893, respectively. Additionally, the three-class classification task exhibited a good discrimination performance with AUCs of 0.893, 0.944, and 0.891 for tasks A, B, and C, respectively.

Two studies in the literature evaluated the performance of radiomics applied to MRI images to predict the type of EOC.

Jian et al. also utilized an MRI-based radiomics model to differentiate between type I and type II OC, recognizing the critical region for differential diagnosis [74].

They reported that the combined model did well in both the internal and external validation cohorts (ROC AUCs of 0.806 and 0.847, respectively). The border zone between the solid and cystic components or the less compact area of the solid component on T2WI Fat Sat has been proven to be the critical region for differentiating type I and type II OC. By identifying the critical region for subtype differentiation, the authors were able to mark out the key area to reduce the sampling error of the intraoperative frozen section, although the application of this model to differentiate between type I and type II OC is yet to be put to test in clinical work.

## 7. Computed Tomography CT

CT is a standard preoperative evaluation of OC patients for disease staging [75].

CT has an accuracy between 70 and 90% for implant detection [75–78] but quite a limited sensitivity (25% to 50%) for implants less than 1 cm, especially in locations such as the bowel or mesentery. Reformatted images in coronal and sagittal planes are very useful for evaluating the subphrenic space [75].

Aside from pre-treatment staging, another aim of the CT exam is the response evaluation to chemotherapy, which, until now, has had some challenges. As it relies only on macroscopic changes such as tumor size—for example, by using the Response Evaluation Criteria in Solid Tumors (RECIST) [79]—CT does not provide a quantitative assessment of disease response to cytotoxic therapy and does not reflect molecular events in tumor-like apoptosis and necrosis.

The presence of carcinomatosis, consisting of many sub-centimeter nodules with poorly defined margins, makes the use of RECIST impossible to achieve. Functional imaging using diffusion-weighted imaging (DWI) has been assessed in this field of research. The use of apparent diffusion coefficient (ADC) histograms may increase the efficacy in the monitoring of responses to treatment in peritoneal carcinomatosis [80].

We report CT studies deploying radiomic analysis and their technical characteristics (Table 3).

**Table 3.** Technical characteristics of reviewed CT studies deploying radiomic analysis.

| Author | Year | Population Number | Radiomic Platform | Radiomics Features | ML Algorithm | Outcome | Study Design |
|---|---|---|---|---|---|---|---|
| Rizzo et al. [81] | 2017 | 101 | IBEX tool (Imaging Biomarker Explorer Software, v. 1.0β) | 516 | Logistic regression | RT, PD12 | Retrospective |
| Wei et al. [82] | 2019 | 142 | ITK-SNAP | 620 | LASSO | 18 months-PFS, 3 years-PFS | Retrospective |
| Himoto et al. [83] | 2019 | 75 | ITK-SNAP-MATLAB | 7 | Cox proportional hazard regressions, LASSO | PFS > 24 weeks | Prospective |
| Vargas et al. [84] | 2017 | 38 | 3D Slicer | 12 | LASSO | CSR, OS | Retrospective |
| Veeraraghavan et al. [85] | 2020 | 75 | Computational environment for radiological research (CERR) | 75 | Support vector machine, Cox proportional hazard regressions | PFS, platinum resistance | Retrospective |
| Lu et al. [86] | 2019 | 364 | TextLAB 2.0 | 657 | Cox proportional hazard regressions | OS | Retrospective |

RT, residual tumor; PD12, 12-month disease progression; LASSO, least absolute shrinkage and selection operator; PFS, progression-free survival; CSR, complete surgical resection; OS, overall survival.

A study of Rizzo et al. extracted the radiomic features from staging CT in OC patients to evaluate if the radiomic features, alone or in combination with clinical data, may be associated with residual tumor (RT) at surgery and to forecast the risk of disease progression within a 12-month duration.

This study showed a good correlation between some radiomic features and prognosis, verifying an AUC of 0.87 [81].

To evaluate an 18-month follow-up in advanced HGSC patients, Wei et al. analyzed radiomic signatures of preoperative CT. According to the timing of their surgery and hospital stay, all patients were divided into cohorts, three in total—the training cohort (TC) and internal validation cohort (IVC) were from one hospital, and the independent external validation cohort (IEVC) was from a different hospital.

The TC showed the most reliable accuracy for forecasting recurrence risk within 18 months and 3 years—82.4% (95% CI, 77.8–87.0%) and 83.4% (95%CI, 77.3–89.6%), respectively.

Patients affected with recurrent OC who were enrolled in prospective immunotherapeutic trials were evaluated by Himoto et al., analyzing their baseline CT scans. The

outcome of interest was sustained clinical benefit, defined as progression-free survival (PFS) of 24 or more weeks. In multivariable analysis, higher energy largest-lesion (indicator of lower intra-tumor heterogeneity; odds ratio (OR) 1.41, (95% CI 1.11–1.81)) and fewer disease sites (OR 1.64, 95% CI 1.19–2.27) were important indicators of strong clinical benefit [83].

Vargas et al. evaluated the associations between clinical outcomes and radiomics-derived inter-site spatial heterogeneity metrics across multiple metastatic lesions on CT in HGSC.

They reported that tumor heterogeneity metrics were most relevant for predicting complete surgical resection (CSR) and survival.

Inter-site texture heterogeneity metrics were evaluated, out of which three measures outlining the differences in texture similarities across sites were associated with shorter OS. No significant association between OS and the remaining nine heterogeneity metrics that averaged the inter-site similarities was observed, though the following features were ranked highest for predicting OS using recursive feature elimination and decision trees: inter-site similarity entropy (SE), similarity cluster prominence (SCP), similarity cluster shade (SCS), correlation, and contrast. Independent evaluation of the data with leave-one-out cross-validation using LASSO regression identified SE for predicting overall survival (OS). The negative weight indicates that smaller values of SE (less heterogeneity) predicted longer OS.

Supervised fuzzy c signifies clustering using the inter-site tumor heterogeneity metrics that outperformed the other texture measures for separating the low-risk group (complete resection, survival of >60 months, no *CCNE1* amplification) and high-risk groups (incomplete surgical resection or had *CCNE1* amplification with overall survival of ≤60 months) with an accuracy of 71%, true positive rate (TPR) of 60%, and true negative rate (TNR) of 86% [84].

Veeraraghavan et al. developed and validated a machine-learning-based integrated marker of HGSC outcomes. An intra- and inter-site radiomics (cluDiss) measure was combined with clinical-genomic variables (iRCG) and compared against conventional (volume and number of sites) and average radiomics (N = 75) for predicting PFS and platinum resistance.

The iRCG model had the best platinum resistance classification accuracy (AUROC 0.78, 95% CI 0.77 to 0.80]. CluDiss was associated with PFS (HR 1.03, 95% CI 1.01 to 1.05) (85).

Lu et al., using machine learning, derived a non-invasive summary-statistic of the primary ovarian tumor based on four descriptors, which they named "Radiomic Prognostic Vector" (RPV).

RPV consists of four radiomic features:

- FD_max_25HUgl;
- GLRLM_SRLGLE_LLL_25HUgl;
- (NGTDM_Contra_HLL_25HUgl;
- FOS_Imedian_LHH (coefficient: 0.250).

All the features appeared to relate to tumor macro-architecture at the 25 Hounsfield unit gray level (and discrete wavelet filters). In biological terms, such features represent the microenvironment of the tumor, the intermixed fibrotic stroma and tumor cells, and the heterogeneity of the tumor.

RPV identified 5% of patients with median overall survival of less than 2 years. It significantly improves established predictive methods and has been validated in two independent, multi-center cohorts.

RPV not only has solid prognostic power (HR 3.83, 95% CI 2.27–6.46) but is also non-invasive and readily accessible, compared to the existing molecular profiles and clinical factors deemed prognostically relevant [86].

## 8. Conclusions

Despite clear potentials, two significant issues are still faced in the adoption of radiomics in routine clinical practice. First, there is no standard protocol for the radiomic feature extraction method for OMs, with the technical variability of imaging instruments being a potential bias. Second, very few authors have provided a prospective external validation on large cohorts for their algorithms. Nevertheless, radiomics analysis represents a potential game changer; imaging can no longer be considered only as a representation of the extent of the patient's disease, but it can also emerge as a powerful tool for the diagnosis and the prediction of clinical and surgical outcomes.

**Author Contributions:** Conceptualization F.A. and G.C. (Gennaro Cormio); methodology V.L.; resources F.A., G.C. (Gerardo Cazzato), and D.L.F.; data curation V.V. and A.S.T.; writing—original draft preparation, F.A., G.C. (Gennaro Cormio), and L.R.; writing—review and editing E.C. and M.M.; visualization G.I.; supervision A.A.K., V.C., G.C. (Gennaro Cormio), and L.R.; project administration F.A. and G.C. (Gerardo Cazzato); funding acquisition: NA. All authors have read and agreed to the published version of the manuscript.

**Funding:** This research received no external funding.

**Institutional Review Board Statement:** Not applicable.

**Informed Consent Statement:** Not applicable.

**Data Availability Statement:** Not applicable.

**Acknowledgments:** We thank the association "ACTO—alleanza contro il tumore ovarico" for supporting the research activity of Francesca Arezzo with the "Adele Leone" grant (outside this study).

**Conflicts of Interest:** The authors declare no conflict of interest.

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
