# Peer review of "Radiomics Analysis in Ovarian Cancer: A Narrative Review"

_applsci, doi:10.3390/app11177833_

Round 1
Reviewer 1 Report
In this manuscript, author has focused on different aspects of Radiomics in ovarian cancer. In ultrosound and CT scan, author has mentioned lot about specific citation, it can be shorten. These look like the work discussion of some paper. It looks nice to see in as table and talk more about the purpose, how far it is beneficial in ovarian cancer and how it can be improved. In the conclusion part, it can be elaborated a little and can conclude with the how the radiomics are in OC and how far it is back logging in OC diagnosis.
Author Response
In this manuscript, author has focused on different aspects of Radiomics in ovarian cancer. In ultrosound and CT scan, author has mentioned lot about specific citation, it can be shorten. These look like the work discussion of some paper. It looks nice to see in as table and talk more about the purpose, how far it is beneficial in ovarian cancer and how it can be improved. In the conclusion part, it can be elaborated a little and can conclude with the how the radiomics are in OC and how far it is back logging in OC diagnosis.
Thank you. We have modified the manuscript summarizing the citations and underlining the purpose. In the conclusions we reported how the radiomics analysis can be improved. We have furthermore modified the conclusions to better underline the aspects requested by you.
Reviewer 2 Report
Title: Radiomics analysis in Ovarian Cancer: a Narrative Review
General review comment: The authors have presented a review on application of radiomics on analysis of ovarian cancer in different imaging modalities such as CT, MRI and ultrasound. This topic is very substantial however, there are some problems which should be taken into account by the authors so as to improve the quality of the paper.
Section of Introduction (line 35 -140):
(1) The manuscript is not written and set well. It looks like a list of papers /notes for presentation. There is too much unnecessary bulleting (itemization) which could be put together in one or two sentences instead. I suggest to remove bulleting from line 51-57, instead combine all items in one/two sentences.
(2) There is unnecessary repetition of the use of conjunction “AND” in the same sentence. Example in line 41, 42, 77 etc. I suggest the author to correct the grammatical errors in the whole manuscript.
(3) I suggest the authors to add a simple paragraph after the introduction which shortly introduces the structure the paper. This will simplify the reading. It should be like..” This paper is structured as follows: Section 2 gives a review on application of radiomics in ultrasound …..etc”
Section of Ultrasound (from line 148 – 195):
(1) The presentation of the reviewed works is poor. There is too much unnecessary information presented from the reviewed work. There are excessive paragraphs for a single reviewed work. Excessive results/metrics presented for a single work. I suggest the authors to summarize each work into one short paragraph containing only necessary information such as motivation of the authors of that particular work, contribution, method/workflow/algorithm used, dataset and results. Please learn more on how to review a paper. When presenting the result choose only one metric (not all) which has the best result. It can be accuracy, specificity, AUC or any.
(2) Generally, there are very few reviewed works which focus on the main idea of the paper (“application of radiomics”). Only four papers [65], [49], [42] and [58]. I suggest the author to add at least 8 more reviewed works on this section
(3) Table 1, 2, and 3 are hanging tables. There is no any point in the sections where they are refereed. Please add statement or paragraph in the corresponding sections which introduces the tables.
(4) Please fit the tables within the margins
(5) All the issues above from item 4-7 are also applicable in the section of MRI (line 196 -249) and computed Tomography (line 251 – 349). Please resolve them as suggested in each item above. Except in computed tomography the reviewed works are enough so no need of adding more papers.
Author Response
Title: Radiomics analysis in Ovarian Cancer: a Narrative Review
General review comment: The authors have presented a review on application of radiomics on analysis of ovarian cancer in different imaging modalities such as CT, MRI and ultrasound. This topic is very substantial however, there are some problems which should be taken into account by the authors so as to improve the quality of the paper.
Section of Introduction (line 35 -140):
- The manuscript is not written and set well. It looks like a list of papers /notes for presentation. There is too much unnecessary bulleting (itemization) which could be put together in one or two sentences instead. I suggest to remove bulleting from line 51-57, instead combine all items in one/two sentences.
Thank you. Modified
- There is unnecessary repetition of the use of conjunction “AND” in the same sentence. Example in line 41, 42, 77 etc. I suggest the author to correct the grammatical errors in the whole manuscript.
Thank you. Deleted.
- I suggest the authors to add a simple paragraph after the introduction which shortly introduces the structure the paper. This will simplify the reading. It should be like..” This paper is structured as follows: Section 2 gives a review on application of radiomics in ultrasound …..etc”
Thank you. Added.
Section of Ultrasound (from line 148 – 195):
- The presentation of the reviewed works is poor. There is too much unnecessary information presented from the reviewed work. There are excessive paragraphs for a single reviewed work. Excessive results/metrics presented for a single work. I suggest the authors to summarize each work into one short paragraph containing only necessary information such as motivation of the authors of that particular work, contribution, method/workflow/algorithm used, dataset and results. Please learn more on how to review a paper. When presenting the result choose only one metric (not all) which has the best result. It can be accuracy, specificity, AUC or any.
Thank you. Summarized
- Generally, there are very few reviewed works which focus on the main idea of the paper (“application of radiomics”). Only four papers [65], [49], [42] and [58]. I suggest the author to add at least 8 more reviewed works on this section
Thank you. To date there are no other works that apply radiomic analysis to ultrasound in the study of ovarian cancer.
- Table 1, 2, and 3 are hanging tables. There is no any point in the sections where they are refereed. Please add statement or paragraph in the corresponding sections which introduces the tables.
Thank you. Amended
- Please fit the tables within the margins
Thank you. Modified
(5) All the issues above from item 4-7 are also applicable in the section of MRI (line 196 -249) and computed Tomography (line 251 – 349). Please resolve them as suggested in each item above. Except in computed tomography the reviewed works are enough so no need of adding more papers.
Thank you. We modified the manuscripet according to your suggestions.

Round 2
Reviewer 2 Report
The authors revised the major problems in paper and the quality of this paper is improved. However, the English in some places is not so well, please ask a native speaker to do proofreading.Author Response
Thank you
The manuscript was revised by a native speaker